# Comparative Genomics and Pathogenicity Analysis of Two Bacterial Symbionts of Entomopathogenic Nematodes: The Role of the GroEL Protein in Virulence

**DOI:** 10.3390/microorganisms10030486

**Published:** 2022-02-22

**Authors:** Abraham Rivera-Ramírez, Rosalba Salgado-Morales, Alfredo Jiménez-Pérez, Rebeca Pérez-Martínez, Blanca Inés García-Gómez, Edgar Dantán-González

**Affiliations:** 1Instituto de Investigaciones en Ciencias Básicas y Aplicadas, Universidad Autónoma del Estado de Morelos, Av. Universidad 1001, Chamilpa, Cuernavaca 62209, Morelos, Mexico; abraham.rivera@uaem.mx; 2Centro de Investigación en Biotecnología, Universidad Autónoma del Estado de Morelos, Av. Universidad 1001, Chamilpa, Cuernavaca 62209, Morelos, Mexico; salgadomoralesr@hotmail.com (R.S.-M.); rebeca_62533@hotmail.com (R.P.-M.); 3Centro de Desarrollo de Productos Bióticos, Instituto Politécnico Nacional, Calle Ceprobi No. 8, San Isidro, Yautepec 62739, Morelos, Mexico; aljimenez@ipn.mx; 4Instituto de Biotecnología, Universidad Nacional Autónoma de México, Apdo. Postal 510-3, Cuernavaca 62250, Morelos, Mexico; blanca.garcia@ibt.unam.mx

**Keywords:** *Xenorhabdus*, *Photorhabdus*, pangenome, pathogenicity, *G. mellonella*, chaperonin

## Abstract

Bacteria of the genera *Xenorhabdus* and *Photorhabdus* are symbionts of entomopathogenic nematodes. Despite their close phylogenetic relationship, they show differences in their pathogenicity and virulence mechanisms in target insects. These differences were explored by the analysis of the pangenome, as it provides a framework for characterizing and defining the gene repertoire. We performed the first pangenome analysis of 91 strains of *Xenorhabdus* and *Photorhabdus*; the analysis showed that the *Photorhabdus* genus has a higher number of genes associated with pathogenicity. However, biological tests showed that whole cells of *X. nematophila* SC 0516 were more virulent than those of *P. luminescens* HIM3 when both were injected into *G. mellonella* larvae. In addition, we cloned and expressed the GroEL proteins of both bacteria, as this protein has been previously indicated to show insecticidal activity in the genus *Xenorhabdus*. Among these proteins, Cpn60-Xn was found to be the most toxic at all concentrations tested, with an LC50 value of 102.34 ng/larva. Sequence analysis suggested that the Cpn60-Xn toxin was homologous to Cpn60-Pl; however, Cpn60-Xn contained thirty-five differentially substituted amino acid residues that could be responsible for its insecticidal activity.

## 1. Introduction

*Xenorhabdus* and *Photorhabdus* are closely related phylogenetic groups belonging to the family Enterobacteriaceae; they are Gram-negative gammaproteobacteria that have evolved to form symbiotic associations with soil entomopathogenic nematodes of the families Steinernematidae and Heterorhabditidae, respectively [1]. These bacteria have a complex life cycle involving a mutualistic symbiotic stage, in which the bacteria become established and colonize the nematode gut, and a pathogenic stage, in which susceptible insects become infected after the nematode enters through natural openings such as respiratory spiracles, the mouth or the anus [2]. Once the bacteria are inside the hemocoel, they actively replicate and release compounds that have the potential to suppress the host immune response as a protective strategy to maintain symbiosis with the nematode. The bacterium enters the stationary phase of its growth cycle, resulting in the production of a wide range of compounds, including toxins, phospholipases, proteases and antibiotics. Degradative enzymes bioconvert the macromolecules of the insect, thus providing a supply of essential nutrients for the nematode reproduction and development, while antibiotic substances inhibit the growth of other microorganisms that might compete for the carcass [3,4]. During the final stages of development, the nematode and bacteria reassociate, and there is a corresponding decrease in nutrients. These processes lead to the differentiation of the nematode from the juvenile stage to the infective juvenile (IJ) stage, in which it no longer feeds on the insect. IJ carrying bacteria in their intestinal tract then emerge from insect carcasses in search of new prey [5,6,7].

Although this nematode–bacteria symbiotic complex shows a high degree of similarity among clades, there are differences in the life cycle itself and in the mechanism of pathogenicity, which involves evasion of the insect immune system and the expression of virulence factors [8]. Some studies have suggested the existence of functionally divergent mechanisms [9,10,11]. One approach to study these differences is based on comparative genomics, which allows the comparison of complete genomes and provides abundant genetic information to elucidate genomic structural landmarks, novel gene repertoires and phylogenetic relationships among different taxa. The fundamental goal of comparative genomics is to achieve pangenome analysis from genomes [12,13]. The pangenome is composed of a core genome, which includes all genes present in all the strains studied, encoding functions related to the basic biology and phenotypes of the species. A second component is an “accessory or dispensable genome,” including genes present in some but not all strains studied as well as strain-specific genes. The dispensable genome is generally associated with nonessential functions, in addition to conferring selective advantages such as adaptability to ecological niches, the ability to colonize new environments, or antibiotic resistance [13,14,15,16]. To date, there have been no pangenome analyses of the entomopathogenic bacteria of genera *Xenorhabdus* and *Photorhabdus.*

In this work, we sequenced and reported the genome of *Xenorhabdus nematophila* SC 0516. In addition, we performed a pangenome analysis with the complete genome sequence data available from 91 different strains of genera *Xenorhabdus* and *Photorhabdus* and determined the phylogenetic position of our strain using the core genome. We also characterized the virulence capacity of *X. nematophila* SC 0516 in *G. mellonella* larvae, comparing it with a strain previously assessed in our laboratory, identified as *Photorhabdus luminescens* HIM3, and we further characterized a protein component with unique properties, GroEL, which is a chaperonin shared by both strains with differential insecticidal activity.

## 2. Materials and Methods

### 2.1. Maintenance of Insects and Nematodes

*Galleria mellonella* was maintained at a temperature of 25 ± 2 °C with a light/dark (LD) photoperiod of 12:12 h and relative humidity (RH) of 70 ± 10% and cultured on an artificial diet. For the isolation of *Heterorhabditis indica*, agricultural soil samples were collected in the State of Morelos, Mexico. Isolation of entomopathogenic nematodes was carried out using *G. mellonella* last-instar larvae as bait. Larvae were monitored daily, dead larvae with symptoms of entomopathogenic nematode infection, such as absence of odor, absence of contaminating organisms and coloration were recovered and disinfected with 5% sodium hypochlorite and rinsed three times with sterile distilled water. Infective juvenile nematodes (IJs) were recovered from white traps 12–14 days after inoculation with IJs. In the case of *Steinernema carpocapsae*, the nematodes were acquired through Kopppert Biological Systems, reproducing the nematodes with the same methodology described above.

### 2.2. Bacterial Growth

Bacterial strains were grown on NBTA medium (nutrient agar supplemented with 0.025 g bromothymol blue and 0.04 g 2,3,5-triphenyltetrazolium chloride per liter). For experimental purposes, a 48-h colony was transferred to 50 mL nutrient broth medium (NB) in a 250 mL Erlenmeyer flask and incubated at 25 ± 1 °C at 120 rpm for 24 h.

### 2.3. Genome Collection

A total of 91 sequences were collected from the annotated genomes of *Xenorhabdus* and *Photorhabdus* available in the open access RefSeq: NCBI Reference Sequence Database, among which 14 were complete, and 77 were drafts in various stages of completion (Appendix A). As a selection criterion, genomes that were highly fragmented (>300 contigs) were excluded. Three complete *Yersinia pestis* genomes were included in the phylogenetic analysis.

### 2.4. Genome Sequencing and Annotation

Genomic DNA from *X. nematophila* SC 0526 was extracted using the ZR Fungal/Bacterial MiniPrepTM Kit (Zymo Research, Irvine, CA, USA), following the manufacturer’s instructions. Subsequently, 5 µg of genomic DNA was sequenced on the Illumina MiSeq platform. Read quality was analyzed with FastQC [17]. Illumina adapter sequences were removed using the ILLUMINACLIP trimming step of Trimmomatic v0.39 software [18]. Low-quality bases were removed from Illumina paired-end reads using the DynamicTrim algorithm of the SolexaQA++ v3.1.7.1 software package with a Phred quality score of Q = 13 [19]. Paired-end reads were assembled de novo using the SPAdes v3.14.1 program with the following options: (i) only run the assembly module (–only-asembler); (ii) reduce the number of mismatches (–careful); and (iii) k-mer lengths between 21 and 71, which generated 248 contigs [20]. Contigs of less than 500 bp in length were discarded, and the remaining contigs were used for a multidraft-based analysis using the genome of the *X. nematophila* YL001 strain via the MeDuSa v1.6 scaffolder [21]. A final assembly polishing step was performed by reassigning the filtered high-quality sequence reads to the ordered scaffolds using BWA and passing the resulting ordered binary alignments to SAMtools for indexing [22]. The indexed alignments were used for analysis by Pilon v1.23 [23]. The draft bacterial genome was automatically annotated using the RAST server version 2.0 [24] (available at https://rast.nmpdr.org/rast.cgi, accessed on 20 July 2021), and 16S rRNA gene sequences were obtained using the RNAmmer server [25] (http://www.cbs.dtu.dk/services/RNAmmer/, accessed on 22 July 2021).

### 2.5. Phylogenomic Analyses

The core genome phylogeny was estimated under the maximum likelihood (ML) criterion using the GET_HOMOLOGUES [26] and GET_PHYLOMARKERS [27] software suites. The get_homologues.pl program was used in combination with compare_clusters.pl to compute a consensus core genome resulting from clustering BLASTP results (with 90% query coverage) with the BDBH (Bi-directional Best Hits), COG (Cluster of Orthologous Groups-triangles) and OrthoMCL (Markov Clustering of orthologs) algorithms implemented in GET_HOMOLOGUES [28]. Domain search was allowed in PFAM for the last runs. A consensus pangenome was similarly computed from COG triangles and OMCL clusters. The pangenome consensus clusters were input into the GET_PHYLOMARKERS pipeline to select alignments with optimal phylogenetic attributes (no significant evidence of recombination, producing tree topologies and branch lengths that did not deviate significantly from the expected distribution of these parameters and showing average branch support values > 0.6). The alignments that passed these filters were concatenated, and an ML phylogeny was estimated with IQ-TREE 1.6.1 [29] using the best-fit model and choosing the phylogeny with the highest probability score of those found among independent searches. Finally, the phylogeny was rooted using the *Y. pestis* genomes. Phylogenetic trees were visualized and edited with FigTree v1.4.3 [30].

### 2.6. Pathogenicity Assays in Galleria mellonella

Pathogenicity bioassays in *G. mellonella* were performed by the direct injection of 10^1^, 10^2^ or 10^3^ CFUs of *X. nematophila* SC 0516 into the hemolymph of fifth-instar larvae of *G. mellonella*. Larvae were selected by weight (250 to 350 mg) in all trials. A final volume of 10 µL of LB containing the bacterial suspension was injected into the hemolymph of individual insects in the last left pro-leg with a 31-gauge insulin needle (BD Medical-Diabetes Care, Holdrege, NE, USA). Equivalent doses of *P. luminescens* HIM3 and *E. coli* DH5α (negative control) were used in all assays. Fifteen larvae were used per dose, and each larva was placed in a 9 cm Petri dish. Experiments were performed four times in independent bioassays (*n* = 60), and survival was assessed at 12, 24, 36 and 48 h post-infection.

### 2.7. Cloning, Expression and Purification of GroEL Proteins

Based on the genomic information obtained from the bacteria, specific primers were designed for the PCR amplification of the groEL sequences of *X. nematophila* SC 0516 (Fw: 5′ CATATGGCAGCTAAAGACGTAAAATTTG 3′ and Rv: 5′ GAATTCACATCATGCCGCCCATTCCAC 3′) and *P. luminescens* HIM3 (Fw: 5′ CATATGGCAGCTAAAGACGTAAAATTTGG 3′ and Rv: 5′ GTCGACTTACATCATGCCGCCCATACCG 3′). The 1.7 kb coding sequences were cloned into the pJET1.2 blunt vector (Thermo Scientific, Vilnius, Lithuania), and the resultant constructs were transformed into *E. coli* DH5α cells following the protocol described by the manufacturer. Subsequently, the 1.7 Kb fragments of both vectors were ligated into the expression vector pET28a, and these constructs were transformed into *E. coli* BL21DE3, thus producing the recombinant protein-producing strains XnGroEL and PlGroEL. These cells were grown at 37 °C and 150 rpm in LB medium containing 50 µg/mL kanamycin until an O.D. of 0.5 was reached, when gene expression was immediately induced by the addition of 1 mM IPTG for 4–5 h. Cells were washed and lysed by sonication, and the cell debris-free supernatant was filtered through a cold agarose-acetic acid-nickel agarose affinity column (Ni-NTA Agarose QIAGEN, Hilden, Germany) following the manufacturer’s recommended protocol.

### 2.8. Evaluation of the Insecticidal Activity of GroEL Proteins

The insecticidal activity of the GroEL proteins from *X. nematophila* SC 0526 (named Cpn60-Xn) and *P. luminescens* HIM3 (named Cpn60-Pl) was evaluated in fifth-instar *G. mellonella* larvae using doses of 100, 200, 500, 1000 and 2000 ng of protein per larva. Injection was performed directly into the insect hemolymph with 0.3 mL 31G X 6 mm Ultra Fine U-100 insulin syringes (BD Medical-Diabetes Care, Holdrege, NE, USA). The final injection volume was 10 µL in all treatments, and BSA (2000 ng) was used as a negative control. The assays were repeated three times using 15 larvae per treatment. The 50% lethal dose of the purified proteins was assessed using semilogarithmic regression, applying the logarithm base 10 of the protein concentrations and using the following formula: y = 42.73x − 35.79.

### 2.9. Sequence and Structure Analysis

The global alignment of the GroEL sequence of *X. nematophila* SC 0516 with that of *P. luminescens* HIM3 was performed with the program Clustal Omega [31]. I-TASSER [32] was used to generate the three-dimensional structure, which was visualized and analyzed with Visual Molecular Dynamics (VMD) software [33,34].

### 2.10. Statistical Analysis

Survival analysis, which is generally a set of methods for analyzing data in which the outcome variable is time to the occurrence of an event of interest (mortality), was performed. Data from the experiments (*n* = 60) were plotted using the Kaplan–Meier estimator, a nonparametric statistic, and a log-rank test was performed to detect significant differences between treatments. To isolate one or more groups that differed from the others, a multiple comparison procedure was carried out using the Holm–Sidak method. In both cases, *p* > 0.005 was the rejection probability.

In the case of semilogarithmic linear regressions, ANCOVA was performed to detect significant differences between the two lines, with a probability of rejection of *p* > 0.005.

Statistical analyses were performed with GraphPad Prism version 8.4.3 (GraphPad Software Inc., San Diego, CA, USA).

## 3. Results

### 3.1. Genomic Features and Assembly Metrics of the SC 0516 Strain

In the present study, bacterial strain SC 0516 was isolated from the hemolymph of *S. carpocapsae*-infected *G. mellonella* larvae. In addition, the bacterial genome of SC 0516 was sequenced and assembled de novo. The draft genome of strain SC 0516 was 4,179,879 bp in length, with 142-fold genome coverage and an approximate 43.5% G+C content. A total of 103 contigs were generated, with an N50 contig length of 73,255 bp and an L50 value of 16 contigs. No plasmids were detected in the analysis.

To functionally analyze the genome of strain SC 0516, the contigs were subjected to subsystem annotation on the RAST server. A total of 4174 protein-coding genes, including 51 tRNAs, 5 rRNAs and 5 noncoding RNAs, were assigned to 336 annotated subsystems, which can be defined as biological processes that are components of metabolism or structural complexes supported by a set of functional roles. Most of these genes were predicted to be involved in the metabolism of amino acids and derivatives (309 genes), proteins (189 genes) and carbohydrates (150 genes).

This whole-genome shotgun project has been deposited in GenBank under accession number JACDOS000000000.1. The version described in this article is the first version.

### 3.2. Pangenome and Phylogenetic Analysis of Xenorhabdus nematophila SC 0516

A pangenome consists of core genes (i.e., genes found in all strains of a genus or species), flexible genes (found in more than one strain but not all), and single genes (found in only one strain, also known as accessory genes). We included 77 drafts and 14 complete genomes of *Xenorhabdus* and *Photorhabdus* bacteria in the analysis (Appendix A). The consensus pangenomic matrix of *Xenorhabdus* and *Photorhabdus* from our dataset consisted of 23,603 gene clusters (Figure 1A). More than half of the complete set of genes constituting the pangenome (15,662 genes, 66%) were found to be uniquely present (referred to as a cloud genome), meaning that each strain contributed an average of 172 new genes to the pangenome. The shell genome (genomes ≥ 50%) and the soft core (genomes = 95%) consisted of 6311 and 1630 gene clusters, respectively (Figure 1A). The analysis showed that the pangenome of both genera has an open form (Figure 1B). This occurs when the number of new gene families continues to increase in a taxonomic lineage in a non-asymptotic trend, regardless of how many new genomes are added to the pangenome. Higher rates of gene gains achieved by horizontal gene transfer (HGT) are characteristic of these lineages [16,35]. The core genome characterized as the set of genes present in all the genomes analyzed contained approximately 348 genes that were present in all 91 genomes studied, representing 1.4% of the pangenome (Figure 1C). However, the core genome presented a negative trend due to the addition of new strains, as the probability of gene sharing between strains decreased as new strains were incorporated into the study sample (Figure 1D). This was consistent with studies revealing a general negative relationship between pangenome size and the proportion of core genes, where larger “open” pangenomes have a lower proportion of core genes [36]. Overall, the *Xenorhabdus–Photorhabdus* pangenome showed a high level of genome variability, with only 1% of its genome being constant. Thus, the remaining 66% of the pangenome is composed of a wide repertoire of genes and molecular functions.

To our knowledge, this is the first pangenome analysis of *Xenorhabdus* and *Photorhabdus* in which the core genomes were examined using BDBH, COG and OMCL strategies. Based on most genes of the core genome grouping according to the NCBI Clusters of Orthologous Groups (COGs) database, many genes are involved in the generation and conversion of energy processes (29.35%), translation, ribosome structure and biogenesis (19.18%), amino acid transport and metabolism (12.40%) and replication, recombination and repair (11.61%), as shown in Figure 2.

The taxonomic position of strain SC 0516 was assessed using a maximum likelihood core genome phylogeny calculated from the 281 highest-scoring markers selected by the GET_PHYLOMARKERS pipeline from the 348 consensus groups calculated by GET_HOMOLOGUES. Three complete genomes of *Y. pestis* were included in the phylogenetic analysis. The core genome phylogeny showed that SC 0516 clustered with *X. nematophila*, as shown in Figure 3, is closely related to *X. nematophila* YL001 (GenBank accession number CP032329.1), isolated from nematodes from a soil sample collected in the locality of Shanxi, China. The strength of the phylogenetic tree obtained using this core genome-based approach resolved clades with maximum support.

In addition, the analysis allowed us to locate genus-specific genes of *Xenorhabdus* and *Photorhabdus,* highlighting those involved in pathogenicity and virulence. Overall, we found that *Xenorhabdus* lacks unique elements related to pathogenicity and virulence. However, the distinctive genes present in the genus included genes associated with tellurium resistance and polyamine transport. On the other hand, *Photorhabdus* presented 29unique elements associated with virulence and pathogenicity related to the type III secretion system, pilus structures and fimbriae (Appendix A). This analysis was consistent with previous observations that *Photorhabdus* possesses genes related to the type III secretion system that are absent in *Xenorhabdus*, and these play an important role in host insect invasion, as well as secretion of toxins and virulence factors [10,11,37,38]. The results showed that, although these genera are phylogenetically related similar lifestyles, they differ drastically in their molecular mechanisms of pathogenicity in the same host. Data indicate that *Xenorhabdus* probably relies on different effectors and secretion systems, which could be reflected in the differences in virulence capacity between the two genera.

### 3.3. Evaluation of Pathogenicity in Galleria mellonella

To evaluate the pathogenicity of *X. nematophila* SC 0516, survival experiments were performed in *G. mellonella* by injecting different doses of colony forming units (CFUs). For a comparative analysis of pathogenicity, we used a bacterial strain previously characterized, identified as *Photorhabdus luminescens* HIM3 [39]. Bioassays showed that both bacteria were pathogenic to *G. mellonella* larvae, as demonstrated by the mortality rate visualized in Kaplan–Meier survival curves, which differed significantly from the control (*E. coli* DH5α) (X^2^ = 457.636, df = 6, and *p* < 0.001), as shown in Figure 4. This analysis revealed that *X. nematophila* SC 0516 was more virulent than *P. luminescens* HIM3 at all doses used (Figure 4). This difference was also illustrated by the median survival times (for which 50% of larvae survive) estimated from the Kaplan–Meier survival analysis. In the case of *X. nematophila* SC 0516, the median times were 26.4 and 24 h for doses of 10^2^ and 10^3^ CFUs, respectively, and 30.4 h for a dose of 10^1^ CFUs. However, *P. luminescens* HIM3 showed median times of 37.2 and 36 h for doses of 10^2^ and 10^3^ CFUs, respectively, and 41 h for a dose of 10^1^ CFUs (Table 1). We also observed differences in the phenotypes of the dead larvae. In the larvae treated with *X. nematophila* SC 0516, a slight darkening of the body was observed after 48 h of treatment, while those treated with *P. luminescens* HIM3 developed a dark reddish color at the same time. No external symptoms were observed in the control larvae (treated with *E. coli* DH5α) (Appendix A).

To explain these differences, we looked for unique virulence- and pathogenicity-related elements present in the genomes of both bacteria. Our analysis showed that *P. luminescens* HIM3 presented a higher number and diversity of pathogenicity- and virulence-associated components than *X. nematophila* SC 0516 in almost all categories (Figure 5). The only unique elements presented by *X. nematophila* SC0516 were the mcf (makes caterpillars floppy) toxin (reported in the literature as a highly insecticidal toxin) and a higher number of genes associated with the type IV secretion system (T4SS) and the type I toxin–antitoxin system (Figure 5). An et al. (2009) compared the gene expression of *Photorhabdus temperata* and *Xenorhabdus koppenhoeferi* in vivo in the insect *Rhizotrogus majalis*, and found that more than 60% of the genes were uniquely induced in one of the two bacteria [40]. However, in *Xenorhabdus*, the mechanisms of toxin delivery to the insect are not completely known. Some mechanisms of secretion through the flagellar apparatus or by vesicular systems of the outer membrane have been suggested [41,42]. The pathogenicity of some Gram-negative bacteria depends on their ability to secrete virulence factors into the mammalian host through the release of outer membrane vesicles (OMVs). Some virulence factors include proteases, hemolysins, phospholipids and lipopolysaccharides [43,44]. The analysis of OMV proteins led to the identification of a 58 kDa GroEL homolog as a major component of a complex characterized as a virulence factor with insecticidal activity in some *Xenorhabdus* species, including *X. nematophila* [45,46]. These factors, termed moonlighting proteins, escaped our pangenome analysis, as was the case of the GroEL chaperonin, and could explain the differences in pathogenicity and virulence found between the two bacteria.

### 3.4. Evaluation of the Insecticidal Activity of GroEL Proteins

The GroEL proteins from both bacteria were cloned and expressed and named Cpn60-Xn and Cpn60-Pl (Appendix A). It should be noted that genomic data showed that the two bacteria possess only a single copy of GroEL. The biological activity of the two purified proteins was evaluated with a direct hemolymph injection. This method introduces the protein directly into the insect, mimicking the release of toxins by bacteria, a phenomenon that occurs shortly after a nematode infects a target insect.

The bioassay results indicated that only the Cpn60-Xn protein killed a high percentage of *G. mellonella* larvae and that the mortality rate depended on the concentration of the protein used. The relationship between larval death and protein concentration was assessed by semilogarithmic linear regression analysis, slope values (m) representing toxicity of 42.73 and 16.87 for Cpn60-Xn and Cpn60-Pl, respectively. The difference between the slopes was highly significant (F = 31.50, DFn = 1, DFd = 26, *p* < 0.001) (Figure 6). The 50% lethal concentration (LC50) of the purified Cpn60-Xn protein was 102.34 ng/larvae. No external symptoms or mortality were observed in control larvae infiltrated with PBS or BSA (2000 ng), or simply punctured. GroEL toxicity has been reported in *Enterobacter aerogenes* (EnGroEL) and some *Xenorhabdus* species. EnGroEL was shown to be a paralytic toxin that ultimately killed cockroaches of the genus *Blatella* when injected into the hemolymph at a minimum dose of 2.7 ± 1.6 ng [47]. A GroEL protein from *X. nematophila* was purified and found to show insecticidal activity against larvae of the cotton bollworm, *Helicoverpa armigera*, after oral administration, but showed no effect when injected into the hemolymph of this insect [48]. Two other proteins from *X. budapestensis* (HIP57) and *X. ehlersii* (XeGroEL) were toxic in *G. mellonella* larvae when injected into the hemolymph, with LC50 values of 206.81 and 0.76 ± 0.08 ng/larva, respectively [49,50]. In *Photorhabdus,* GroEL was reported as a protein component of an 860 kDa complex found in the toxic fraction of *P. luminescens* W-14 extract [51]. However, as no direct evidence supporting this finding reported, this study provides the first evaluation of the activity of this protein.

A global alignment of the two protein sequences showed that they share 89.86% identity for 548 residues. Cpn60-Xn showed 35 substitutions relative to Cpn60-Pl, which were distributed throughout the protein. A three-dimensional homology-generated model of Cpn60-Xn showed that 13 of these substitutions were located in the apical domain, 2 in the intermediate domain, and 20 in the equatorial domain (Appendix A). These point substitutions could be responsible for conferring the greater toxicity of Cpn60-Xn relative to Cpn60-Pl.

## 4. Conclusions

In this study, we sequenced and reported the genome of *X. nematophila* strain SC0516, which allowed us to perform the first robust pangenomic analysis of 91 genomes of *Xenorhabdus* and *Photorhabdus*, using three different algorithms. The results revealed that the differences between these two genera mainly involved genes belonging to the type III secretion systems and related to the fimbria and pilus, which are absent in *Xenorhabdus.* Despite this, data from biological experiments related to pathogenicity and virulence showed that both *X. nematophila* SC0516 and *P. luminescens* HIM3 strains were pathogenic. However, the former was more virulent at all concentrations tested, indicating differences in the mechanism of pathogenicity. In addition, the GroEL protein (reported as a virulence factor in *Xenorhabdus*) from *X. nematophila* SC0516 (Cpn60-Xn) showed significantly more insecticidal activity than that from *P. luminescens* HIM3 (Cpn60-Pl), at all concentrations when injected in *G. mellonella* larvae. Analysis of GroEL sequences from both bacteria showed that this protein was highly conserved in *X. nematophila* SC 0516 and *P. luminescens* HIM3, although the former presented 13 specific differences that could explain the difference observed in insecticidal activity.

## Figures and Tables

**Figure 1 microorganisms-10-00486-f001:**
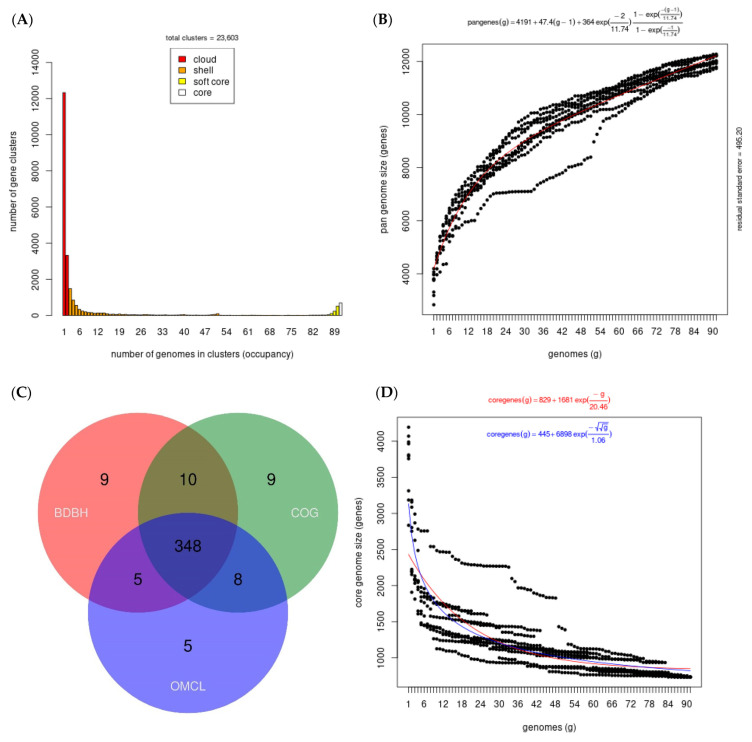
(**A**) Bar plot of the frequency of genes within the 91 genomes included in this analysis. Genes present in a single genome are lineage-specific, while at the opposite end of the scale, genes found in all 91 genomes represent the core genome. (**B**) Plot of the estimated pangenome sizes of *Xenorhabdus–Photorhabdus* fitted with the Tettelin function. (**C**) Venn diagram of the core genome generated by the bidirectional best-hit (BDBH), Cluster of Orthologous Groups (COG), and Markov Clustering of orthologs (OMCL) algorithms. (**D**) Plot of the estimated core genome sizes *of Xenorhabdus–Photorhabdus*.

**Figure 2 microorganisms-10-00486-f002:**
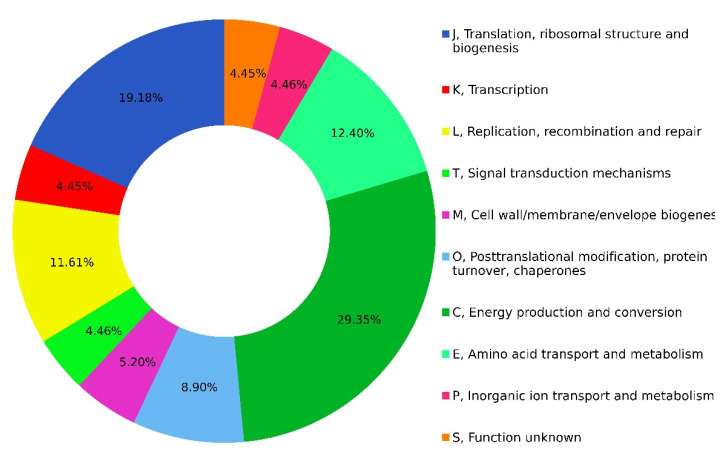
COG functional classification of genes belonging to the *Xenorhabdus–Photorhabdus* core genome. COG database annotation was used to classify the 348 members belonging to the core genome. The abundance of specific gene classes is shown.

**Figure 3 microorganisms-10-00486-f003:**
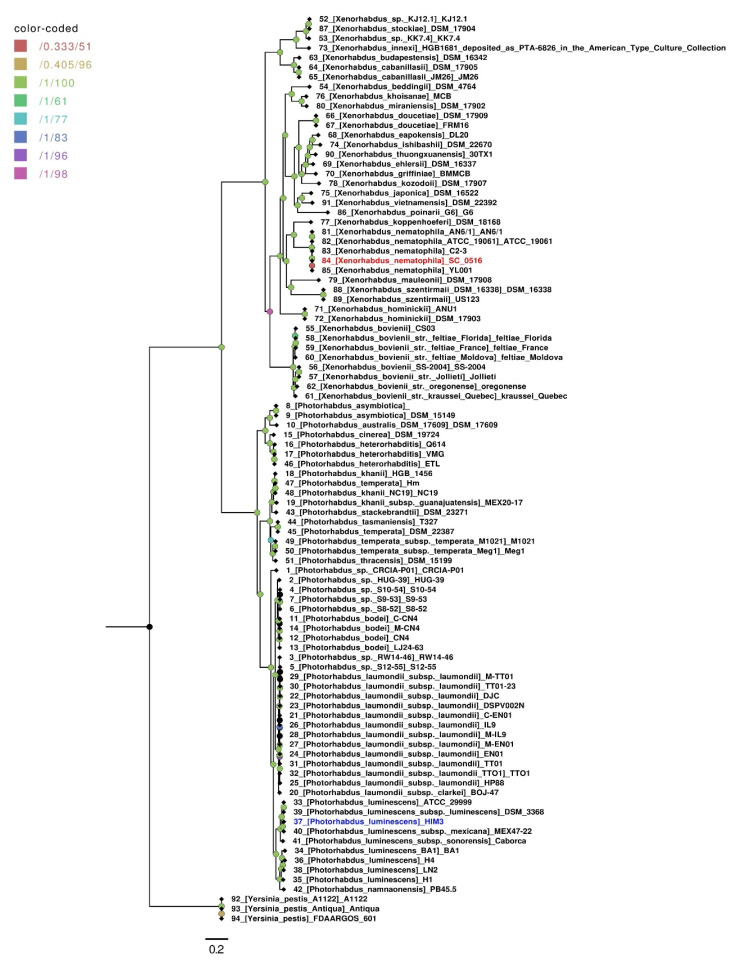
Maximum likelihood (ML) core genome phylogeny. The ML tree is based on 281 consensus protein-coding genes defined by the GET_HOMOLOGES package and filtered through GET_PHYLOMARKERS, selected for their optimal phylogenetic attributes. Nodal support values are color-coded as shown in the legend. The first value corresponds to approximate Bayesian support values, whereas the second one corresponds to ultrafast bootstrap values, as implemented in IQTREE. The scale represents the number of expected substitutions per site under the best-fit GTR2+F+R3 (binary) model. The phylogeny corresponds to the tree with the highest score (lnL = −3,045,265.731) found among 10 independent searches in IQTREE.

**Figure 4 microorganisms-10-00486-f004:**
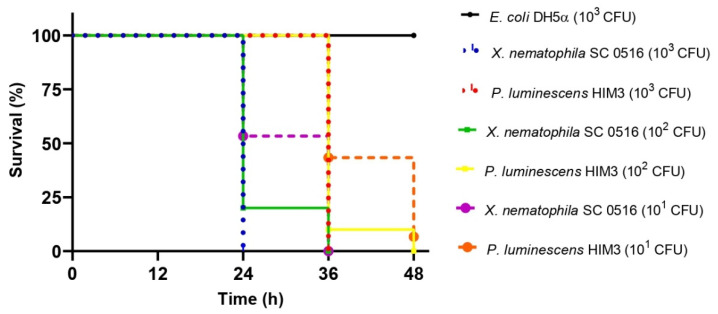
Kaplan–Meier survival curves of *G. mellonella* larvae after direct injection with *X. nematophila* SC 0516. Experiments were performed four times in independent bioassays, and survival was assessed at 12, 24, 36 and 48 h after infection. Differences in survival (*n* = 60) were calculated using the log-rank test X^2^ = 457.636, df = 6, *p* < 0.001.

**Figure 5 microorganisms-10-00486-f005:**
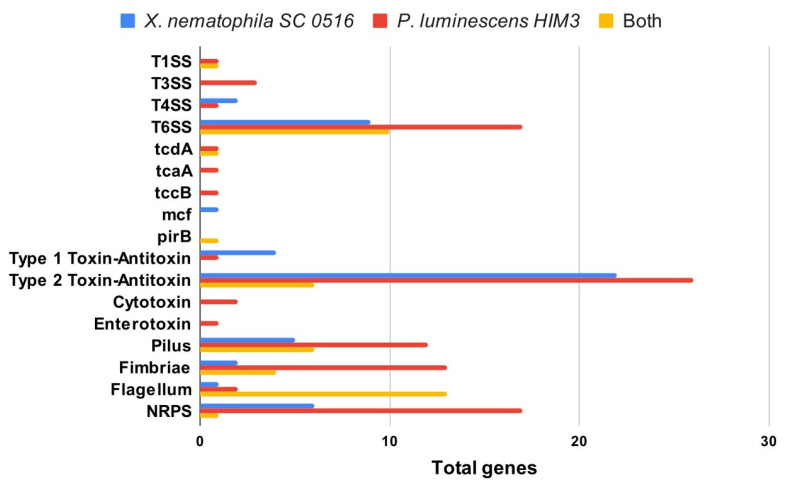
Unique and shared genes between *X. nematophila* SC 0516 and *P. luminescens* HIM3. The x-axis indicates the total number of genes present in one or both bacteria, and the y-axis indicates the types of genes associated with virulence; Type Secretion System (TSS), Toxin Complex (tc), Makes Caterpillars Floppy (mcf), *Photorhabdus* Insect-Related Toxins (Pir), and Non-Ribosomal Peptide Synthetases (NRPS).

**Figure 6 microorganisms-10-00486-f006:**
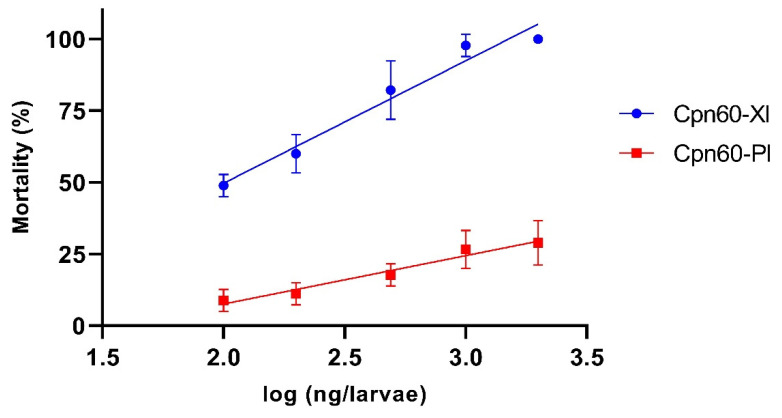
Insecticidal activity of GroEL from *X. nematophila* SC 0516 (Cpn60-Xn) and *P. luminescens* HIM3 (Cpn60-Pl) against *G. mellonella* larvae. Plot of percent mortality against the protein concentration. Each dot in the plot represents the average of three individual experiments ± standard error.

**Table 1 microorganisms-10-00486-t001:** Survival time (h) of *G. mellonella* larvae injected with different doses of colony forming units (CFUs) of *X. nematophila* SC 0516 and *P. luminescens* HIM3.

Strain	10^1^ CFU	10^2^ CFU	10^3^ CFU
*E. coli* DH5α	-		48 A
*P. luminescens* HIM3	41 ± 0.771 A	37.2 ± 0.469 A	36 B
*X. nematophila* SC 0516	30. 4 ± 0.779 B	26.4 ± 0.625 B	24 C

Log-rank test: 457.636, df: 6, *p* < 0.001. Survival time on the same column followed by the same letter is not significantly different (Holm–Sidak test, *p* > 0.05).

## Data Availability

All data associated with this manuscript are given in the manuscript, such as in the Supplementary Material; the entire genome of *X. nematophila* SC0516 was sequenced and deposited in GenBank (NCBI) with the assigned accession number: JACDOS000000000.1 (https://www.ncbi.nlm.nih.gov/nuccore/JACDOS000000000.1/, accessed on 30 December 2021).

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
