# Peer review of "Comparative Genomics and Pathogenicity Analysis of Two Bacterial Symbionts of Entomopathogenic Nematodes: The Role of the GroEL Protein in Virulence"

_microorganisms, 2022, doi:10.3390/microorganisms10030486_

Round 1
Reviewer 1 Report
Generally well-written article, covering an important topic as pathogenicity of entomopathogenic symbionts – Xenorhabdus or Photorhabdus bacteria. Authors analyzed 91 strains of each genera. The authors showed that the Photorhabdus has a higher number of genes associated with pathogenicity. Interestingly, with simultaneous exposure of G. mellonella to cells of X. nematophila SC0516 which is associated with S. carpocapsae was more virulent than P. luminescens HIM3 associated with Heterorhabditis spp. The results are presented in a clear and transparent manner. The analysis allowed to locate genes of Xenorhabdus or Photorhabdus involved in pathogenicity and virulence. The conducted research may constitute a good basis for a better understanding of substances secreted by bacteria that allow the insect's immune system to overcome.
Minor comments:
In Table 1. – If times in each column were compared, then in the first and second columns, statistical differences should be marked with letters A and B, not B and C.
I suggest increasing the legend and markers on the figure 1 and figure 2, so that they are better visible.
Author Response
Response to Reviewer 1
Thank you very much for these kind words. Your comments motivate us to continue in our projects. We have worked hard to improve the structure and writing of the manuscript.
- In Table 1. – If times in each column were compared, then in the first and second columns, statistical differences should be marked with letters A and B, not B and C.
Response: We appreciate the observation; all questions have been satisfactorily answered.
- I suggest increasing the legend and markers on the figure 1 and figure 2, so that they are better visible.
Response:
Thank you for these observations. The suggested correction has been made
Reviewer 2 Report
The study provides new information on the genome organization of two bacteria of interest for exploitation in biological control. The manuscript, however, shows a first comparative overview and the different aspects studied are only superficially treated. Although it lacks focus on any specific issue, the data provided may be of interest for other scientists working in the same research field. It is suggested to adopt a more concise style, as many sentences are too long and difficult to read. Some minor changes are shown on the attached, edited document. The Authors are also requested to emend the manuscript as follows:
lines 76-78 = this sentence is not clear, it may induce the reader to think that SC516 and HUM3 are the same bacteria.
line 85 = give more information on the nematode species and strains, including geographic origin of bacteria, insect and nematodes used for the original isolation, and collection or deposit locations or names.
line 195 = give the species name before "infected larvae".
Figure 2 = add, on the plot, the percent or number of genes per class.
Figure 4 = improve the image resolution.
line 174 = the asterisk, as a multiplication symbol, is not needed.
lines 234, 242, 393 and others = always show the genus or species names in italics.
lines 300-301 = use superscripts.
line 311 = is this x2 ?
In legend of Fig. S1 = erase "shows" and replace with ":"
In legend of Fig. S3 = add "shows" after "(:)" and, if the substitutions are indicated in red, add "(red)" after "substitutions".
Eliminate the SNAS Editing Certificate from the Supplementary files, it's not appropriate to show it in the final article. The manuscript may be accepted after the requested amendments are applied.

Author Response
Response to Reviewer 2
We appreciate the positive feedback from the reviewer. We have answered each of your points below.
- lines 76-78 = this sentence is not clear, it may induce the reader to think that SC516 and HUM3 are the same bacteria.
Response:
We appreciate the observation, and the paragraph was rewritten. (Line 75-78)
New paragraph:
We also characterized the virulence capacity of X. nematophila SC 0516 in G.mellonella larvae, comparing it with a strain previously assessed in our laboratory, identified as Photorhabdus luminescens HIM3, and we further characterized a protein component with unique properties
- line 85 = give more information on the nematode species and strains, including geographic origin of bacteria, insect and nematodes used for the original isolation, and collection or deposit locations or names.
Response:
We appreciate the observation, and the paragraph was rewritten, and the requested data has been added. (Line 85-95)
New paragraph:
For the isolation of Heterorhabditis indica, agricultural soil samples were collected in the State of Morelos, Mexico. Isolation of entomopathogenic nematodes was carried out using G. mellonella last-instar larvae as bait. Larvae were monitored daily, dead larvae with symptoms of entomopathogenic nematode infection, such as absence of odor, absence of contaminating organisms and coloration were recovered and disinfected with 5% sodium hypochlorite and rinsed 3 times with sterile distilled water. Infective juvenile nematodes (IJs) were recovered from white traps 12-14 days after inoculation with Ijs. In the case of Steinernema carpocapsae, the nematodes were ac-quired through Kopppert Biological Systems, reproducing the nematodes with the same methodology described above.
- line 195 = give the species name before "infected larvae".
Response:
The suggested correction has been made. We add Galleria mellonella (Line 200-201)
New paragraph:
In the present study, bacterial strain SC 0516 was isolated from the hemolymph of S. carpocapsae infected G. mellonella larvae.
- Figure 2 = add, on the plot, the percent or number of genes per class.
Response:
Thank you for recommendation. The percentages of genes have been added
- Figure 4 = improve the image resolution.
Response:
Thank you for this excellent observation. Image resolution was improved
- line 174 = the asterisk, as a multiplication symbol, is not needed.
Response:
Thank you for these observations, the asterisk has been removed
- lines 234, 242, 393 and others = always show the genus or species names in italics.
Response:
We appreciate the observation; all questions have been satisfactorily answered.
- lines 300-301 = use superscripts.
Response:
We appreciate the observation; all questions have been satisfactorily answered. (Line 299, 303 and 304)
- line 311 = is this x2 ?
Response:
We appreciate the observation. The correction has been made (line 314):
New paragraph:
Differences in survival (n=60) were calculated using the log-rank test X2= 457.636, df= 6, p <0.001.
- In legend of Fig. S1 = erase "shows" and replace with ":"
Response:
We appreciate the observation. The correction has been made (Supplementary informational):
New paragraph:
Figure S1. Phenotypes of G. mellonella larvae after injection X. nematophila SC 0516. From left to right: the phenotype of the fifth instar larva of G. mellonella infected with X. nematophila SC 0516, P. luminescens HIM3 and the E. coli DH5a (negative control) at 48 hours post infection.
- In legend of Fig. S3 = add "shows" after "(:)" and, if the substitutions are indicated in red, add "(red)" after "substitutions".
Response:
We appreciate the observation. The correction has been made (Supplementary informational):
New paragraph:
Figure S3. (A) Global alignment of the Cpn60-Xl and Cpn60-Pl sequences. The asterisks (*) below the sequences represent identical amino acids, (:) shows amino acids with similar physicochemical properties and (.) amino acids with different properties. (B) Mapping on the 3D structure of the GroEL protein of the 35 different substitutions (red) between Cpn60-Xl and Cpn60-Pl.
- Eliminate the SNAS Editing Certificate from the Supplementary files, it's not appropriate to show it in the final article. The manuscript may be accepted after the requested amendments are applied.
Response:
We appreciate your observation. The SNAS Certificate has been deleted.
The authors want to thank the comments of the reviewer that improve substantially the manuscript.